# Cross-Task Knowledge Transfer for Semi-supervised Joint 3D Grounding and Captioning

## ABSTRACT

3D visual grounding is a fundamental yet important task in multimedia understanding, which aims to locate a specific object in a complicated 3D scene semantically according to a text description. However, this task requires a large number of annotations of labeled text-object pairs for training, so the scarcity of annotated data has been a key obstacle in this task. To this end, this paper makes the first attempt to introduce and address a new semi-supervised setting, where only a few text-object labels are provided during training. Considering most scene data has no annotation, we explore a new solution for unlabeled 3D grounding by additionally training and transferring knowledge from a correlated task, *i.e.*, 3D captioning. Our main insight is that 3D grounding and captioning are complementary and can be iteratively trained with unlabeled data to provide object and text contexts for each other with pseudo-label learning. Specifically, we propose a novel 3D Cross-Task Teacher-Student Framework (3D-CTTSF) for joint 3D grounding and captioning in the semi-supervised setting, where each branch contains parallel grounding and captioning modules. We first pre-train the two modules of the teacher branch with limited labeled data for warm-up. Then, we train the student branch to mimic the ability of the teacher model and iteratively update both branches with the unlabeled data. In particular, we transfer the learned knowledge between the grounding and captioning modules across two branches to generate and refine the pseudo-labels of unlabeled data for providing reliable supervision. To further improve the quality of the pseudo-labels, we design a cross-task pseudo-label generation scheme, filtering low-quality pseudo-labels at the detection, captioning, and grounding levels, respectively. Experimental results on various datasets show competitive performances in both tasks compared to previous fully- and weakly-supervised methods, demonstrating the proposed 3D-CTTSF can serve as an effective solution to overcome the data scarcity issue.

## CCS CONCEPTS

• **Information systems → Novelty in information retrieval**.

## KEYWORDS

3D visual grounding, semi-supervised learning, teacher-student framework, cross-task knowledge transfer, pseudo-label generation

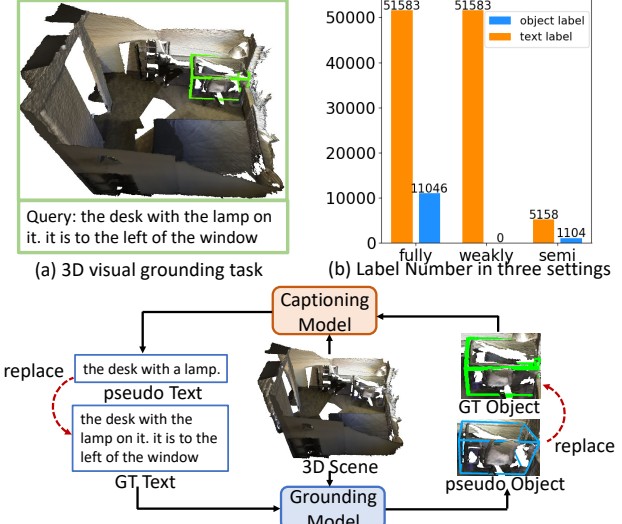

(a) 3D visual grounding task

Query: the desk with the lamp on it. it is to the left of the window

(b) Label Number in three settings

(c) Cross-task knowledge transfer handling labeled and unlabeled data

**Figure 1: We tackle the 3D grounding task in a challenging semi-supervised setting with limited annotations. We propose cross-task knowledge transfer to learn the models with labeled and unlabeled data.**

## 1 INTRODUCTION

3D visual grounding (3DVG) [1, 8] is a fundamental yet important task in multimedia understanding, which has recently received increasing attention due to its wide range of applications, such as robotic navigation and Augmented Reality / Virtual Reality systems. As shown in Figure 1 (a), 3DVG aims to locate the target object in a 3D point cloud scene based on a given free-form query text description. Since 3D scenes generally contain complicated background objects and spatial relationships, this task needs to not only model the complex multi-modal interactions among features of the point cloud and language, but also capture complicated context information for their semantic reasoning.

Existing methods for the 3DVG task can be generally categorized into two types based on task settings: fully-supervised [1, 7–9, 20, 22, 45, 49, 51, 52] and weakly-supervised methods [44, 48]. As shown in Figure 1 (b), under the fully-supervised setting, a large number of bounding boxes for numerous objects in the 3D point cloud scenes, along with their descriptive texts, are required to be annotated as labels to provide reliable supervision. For example, the widely used ScanRefer dataset [8] contains 806 scenes with 51,583 matched object-text labels. However, manually annotating these object-text pairs is very time-consuming and labor-intensive. To alleviate this problem, the weakly-supervised methods [44, 48] are proposed to only use text labels for grounding without relying on any object bounding box annotations. Although they slightly reduce

the annotation cost, tens of thousands of language descriptions still require significant manual effort for annotation.

Based on the above observation, in this paper, we focus on how to learn a 3D visual grounding model with as few annotations as possible. In particular, we introduce a new semi-supervised setting for the 3DVG task, where only a small number of object-text pair labels (about 10%) in 3D scenes need to be manually annotated while the vast majority of scenes have no manual annotations. Since most scene data lack annotations, we explore the idea of cross-task assistance and knowledge transfer from a related task, *i.e.*, the 3D captioning task that aims to generate a contextual sentence description corresponding to a given object within the 3D scene. The motivation is illustrated in Figure 1 (c), where we observe that both tasks are complementary to each other as grounding can provide detected objects for feeding captioning while captioning can generate reliable sentences to guide the grounding process. However, it is non-trivial to directly build a collaborative optimization scheme for these two tasks: (1) **Efficient model design**. Although directly employing a pre-trained 3D captioning model to provide guidance for unlabeled 3D grounding does work, this defeats our purpose of only using a small number of annotations for training, and introduces large memory costs. How to efficiently and effectively train both 3D grounding and captioning models *from scratch* with annotation-limited grounding data is crucial. (2) **Framework for addressing the semi-supervised setting.** In the semi-supervised setting, the limited annotations cannot provide strong supervision to generalize our grounding and captioning models. How to design a framework that can both learn the capability of addressing the task from the limited annotations and fine-tune the models for better generalization with the large amount of unlabeled data is also essential. (3) **Cross-task knowledge transfer.** Instead of individually training the grounding and captioning models, it is important to have them cooperate to provide additional contexts for each other during the learning. Especially when learning with unlabeled data, how to explore their task-specific knowledge to provide rectification on pseudo-labels for each other for better updating the models is challenging.

To address the above issues, we propose a novel approach, called 3D Cross-Task Teacher-Student Framework (3D-CTTSF), to jointly learn both 3D grounding and captioning tasks in a single backbone model with two different task-specific heads. To handle the challenging semi-supervised learning, 3D-CTTSF designs two branches sharing the same joint backbone to build a teacher-student framework. Specifically, 3D-CTTSF first pre-trains the two task-specific heads of the teacher branch with limited labeled data to acquire a certain level of proficiency in tackling the tasks for warm-up. Then, it trains the student branch to mimic the ability of the teacher model and utilizes the pseudo-labels of unlabeled data predicted by the teacher model for updating the student branch. To make two task-specific heads cooperate to provide learning guidance for each other, we transfer the learned knowledge between the grounding and captioning modules across two branches. This transfer helps rectify and refine the pseudo-labels of unlabeled data for providing reliable supervision. To further improve the pseudo-label quality, we design a cross-task pseudo-label generation scheme, filtering out low-quality pseudo-labels at the detection, captioning, and grounding levels, respectively. Through pseudo-label training

and knowledge transfer, we address not only the semi-supervised 3D visual grounding task but also the semi-supervised 3D dense captioning task. During the inference, we directly feed the data into the student model for prediction.

Our main contributions can be summarized as follows:

- To the best of our knowledge, we are the first to address the 3D visual grounding task in a challenging semi-supervised setting. To handle the limited annotation, we propose a novel paradigm that explores cross-task assistance from a complementary task of 3D captioning.
- To tackle semi-supervised learning, we develop a teacher-student framework based on a joint grounding and captioning backbone to train with labeled data and update with pseudo-labels of the unlabeled data. Cross-task knowledge transfer and pseudo-label filtering modules are further developed to aid learning from unannotated scenes.
- Extensive experiments are conducted on various datasets of both grounding and captioning. Results show that our proposed 3D-CTTSF bridges the gap between semi-supervised learning and fully-supervised methods.

## 2 RELATED WORK

**3D Visual Grounding.** Existing 3D visual grounding methods can be generally classified into two-stage and one-stage approaches based on their model designs. Two-stage methods [1, 3, 7, 8, 12, 16, 18, 20, 21, 34, 49, 51, 52] first utilize pre-trained 3D detectors [17, 33, 54] or segmentors [15] to generate a large number of candidate proposals, and then fuse proposal features with textual features to select the object that best matches the given text. One-stage methods [22, 29, 45] directly guide object detection using textual cues, making it easier to locate the object related to the query text. However, almost all of them are developed in the fully-supervised setting where a large number of object-text annotations are required. Although a few weakly-supervised methods [44, 48] are proposed to only use text labels for grounding without relying on any object bounding box annotations, a large number of language descriptions still need to be annotated. Different from them, we are the first to attempt to solve this task with a semi-supervised approach. Although D3net [9] claims applicability in a semi-supervised setting, they still utilize all ScanRefer data as labeled data and only employ some unannotated objects from Scannet as unlabeled data. This setting is much simpler than ours but has not demonstrated superior performance.

**3D Dense Captioning.** Another focal point in 3D multi-modal scene understanding is 3D dense captioning [11]. This task aims to generate natural language descriptions for each object within a scene. Similar to visual grounding, methodologies in 3D dense captioning can be classified into two categories: two-stage and one-stage methods. Two-stage method [11, 24, 43, 50] follow "detect-then-describe" pipeline. They first detect many proposals and then generate captions for each of them. The one stage method [10] directly creates object queries from the scene. The query then be decoded for detection and captioning.

**Semi-Supervised Learning.** Semi-Supervised Learning (SSL) has made significant progress in various tasks, such as image classification [5, 6, 19, 36], object detection [23, 26, 27, 30, 37, 53] and

referring expression comprehension [38]. The methods of semi-supervised learning can be roughly divided into consistency based methods [5, 6, 23, 36] and pseudo-label based methods [2, 26, 27, 30]. Consistency-based methods utilize consistency regularization, which forces both the teacher and student models to produce the same predictions from differently augmented inputs. On the other hand, pseudo-label based methods supervise the training of the student model based on pseudo-labels generated by a reliable teacher model. Many techniques have been proven effective in semi-supervised learning and have thus been widely applied, such as EMA [39] update for teacher, asymmetric data augmentation [35, 46], and pseudo-label filtering [36].

## 3 THE PROPOSED METHOD

### 3.1 Overview

**Task definition.** Semi-supervised 3D grounding and captioning aim to utilize a single pipeline to jointly locate the text-related object and generate the object-guided sentence with only a few annotations. To be specific, the given semi-supervised dataset includes a small set of labeled data $D_l = \{\mathbf{P}_i^l, \mathbf{L}_i^l, \mathbf{B}_i^l\}_{i=1}^{N_s}$ and a large set of unlabeled data $D_u = \{\mathbf{P}_i^u\}_{i=1}^{N_u}$, where $\mathbf{P}, \mathbf{L}, \mathbf{B}$ denote point cloud of a 3D scene, language description and localization of objects, respectively. $N_s$ and $N_u$ are the numbers of labeled data and unlabeled data. In practice, they often meet $N_s \ll N_u$.

**Overall pipeline.** To address the joint tasks in the challenging semi-supervised setting, we propose a novel 3D Cross-Task Teacher-Student Framework (3D-CTTSF). Both the teacher and student branches share a similar architecture with a detection module and parallel grounding and captioning modules. As shown in Figure 2, the overall training process consists of two stages. In the first stage, we pre-train the grounding and captioning modules of the teacher model with a few annotated data for warm-up. By doing so, we obtain a teacher model with a certain capability to perform both grounding and captioning tasks. Then, we copy the parameters of the teacher model to the student model in preparation for the next stage. In the second stage, we train the student branches with both labeled data and unlabeled data. Unlabeled data is supervised using pseudo-labels, while labeled data is supervised using the ground truth. Specifically, we utilize the teacher model to generate the pseudo object proposals and corresponding pseudo textual descriptions for the unlabeled 3D scenes. To improve the quality of pseudo-labels, we introduce a cross-task pseudo-label generation scheme, filtering out low-quality pseudo-labels at the detection, captioning, and grounding levels, respectively. Then, we apply these pseudo-labels to train the student model while the teacher model also updates its parameters from the student model using exponential moving average (EMA) [39] strategy. During the inference, we directly feed the test data into the student model for prediction.

### 3.2 Joint Grounding and Captioning Backbone

**Backbone Model.** We follow previous work [7] to design our basic joint grounding and captioning backbone model, which consists of three main modules: an object detection module $O$, a grounding module $\mathcal{G}$, and a captioning module $C$. In the object detection module, the input 3D scene point cloud is processed through a

Votenet [33] to obtain initial object proposals. For the captioning module, it selects the proposal with the maximum Intersection over Union (IoU) value with the input bbox to generate a caption in training, and directly generates captions for all proposals in testing. For the grounding module, it predicts confidence scores for all proposals based on the input text.

**Design of IoU estimation module.** To adapt the backbone to semi-supervised setting, we require the model to handle the pseudo-labels of unlabeled data. To evaluate the quality of pseudo-labels, we design an IoU estimation module upon the detection module to predict the IoU values between each proposal and the ground truth bounding box. To fully capture the geometric information of proposals, inspired by BRNet [13], we first sample two representative points in each of the six directions (up, down, left, right, front, back) around the given proposal. These 12 number of representative points are evenly distributed along the axes from the bounding box center to the center of each face. By aggregating the contextual features of seed points around these representative points, we obtain and fuse the features of all representative points to predict the IoU value of the proposal.

### 3.3 Teacher-Student Framework for Semi-supervised Setting

After introducing the basic backbone of joint tasks, we then illustrate how we build the detailed framework for addressing the challenging semi-supervised setting.

**How to tackle the semi-supervised learning?** In the semi-supervised setting, we only have access to very few paired object-query samples while the others are all unlabeled. A general way to address this setting is to utilize a teacher-student framework. Specifically, the teacher model is initially trained on a limited number of annotated samples for several epochs to acquire a certain level of proficiency in the given task. Subsequently, the learned teacher model is utilized to assign pseudo-labels to the other unlabeled data. The unlabeled data, now augmented with pseudo-labels, can be further utilized alongside labeled data to train the student model.

**Design of the teacher-student framework.** Our teacher-student framework consists of a teacher model and a student model, which share the same architecture of the joint task backbone $[O, C, \mathcal{G}]$. For clarity, variables and modules associated with teacher and student models are denoted with superscripts "$t$" and "$s$", respectively.

The first stage, also called the warm-up stage, is to pre-train the teacher model with a few annotated samples. Specifically, we train the teacher model using annotated dataset $D_l = \{\mathbf{P}_i^l, \mathbf{L}_i^l, \mathbf{B}_i^l\}_{i=1}^{N_s}$, while the ground truth labels of text $\mathbf{L}^l$ and bounding box $\mathbf{B}^l$ pair are utilized to supervise the training. By denoting the loss function as $\mathcal{L}_l$ (illustrated in Sec.3.6), the parameters of the teacher model $[O_t, C_t, \mathcal{G}_t]$ are updated through the following gradient descent:

$$[O_t, C_t, \mathcal{G}_t]^j = [O_t, C_t, \mathcal{G}_t]^{j-1} + \gamma \frac{\partial \mathcal{L}_l}{\partial [O_t, C_t, \mathcal{G}_t]^{j-1}}, \quad (1)$$

where $\gamma$ is the learning rate and $j$ is the training step index. After training in warm-up stage for $E_1$ epochs, we copy the parameters of the teacher model to the student model and start the training for the semi-supervised stage for $E_2$ epochs.

**Figure 2: The pipeline of our proposed method 3D-CTTSF. The 3D-CTTSF consists of two stages. (a) In the first stage, a small amount of annotated data is used to train the teacher model. (b) In the second stage, the teacher model generates pseudo-labels for object-query pairs for unlabeled scene point clouds, which are utilized in training the student model.**

In the second stage, we update the weights of the student model by using pseudo-label learning with unannotated samples. To be specific, the unlabeled dataset $D_u$ is first fed into the teacher model to generate the pseudo-labels, and then the pseudo-labels are utilized to train the student model. In particular, we employ a training strategy of strong data augmentation for the student model and weak data augmentation for the teacher model. This strategy enables the teacher model to have a performance advantage over the student model, allowing the student to learn from the teacher. Specifically, for the input scene point cloud of the teacher model, we only apply point cloud resampling as weak data augmentation. For the student model, in addition to resampling, we employ a combination of random flips along the y-axis, random translations, random size scaling, and random slight rotations as strong data augmentation. Note that random flipping may change the relative positions of objects, so corresponding modifications to the textual descriptions are required. We did not adopt flipping along the x-axis due to the relatively complex expression of front-back relationships in natural language. For flipping along the y-axis, we can simply swap the words "left" and "right" in the textual description. After asymmetric data augmentation, the input point clouds of the teacher and student models are denoted as $\mathbf{P}_t$ and $\mathbf{P}_s$, respectively.

Next, $\mathbf{P}_t$ will be input into the teacher model for generating corresponding pseudo-labels, and the obtained pseudo-labels will be used for training the student model. The teacher model is also updated through an EMA strategy illustrated in the latter section. To obtain high-quality pseudo-labels for better training the whole teacher-student framework, we design a cross-task pseudo label generation for supervising different modules in the next section.

## 3.4 Cross-Task Pseudo Label Generation

Due to the limited number of labeled data and the complexity of the joint task, the performance of the teacher model does not guarantee the generation of high-quality pseudo-labels. However, low-quality pseudo-labels will reduce the learning efficiency and effectiveness of the student model. This necessitates a filtering process of the pseudo-labels generated by the teacher model to provide true-positive labels for learning student model. We consider improving the quality of pseudo-labels across tasks in three parts: detection module, captioning module, and grounding module.

**Pseudo label of detection module.** As shown in Figure 2 (b), the scene point cloud $\mathbf{P}_t$ undergoes weak data augmentation and is then input into the detection module $O_t$ of the teacher model.

This module detects $K_0$ objects and outputs their bounding box $\mathbf{b}_t$, objectness score $o_t$, classification score $\mathbf{c}_t$, and IoU value $i_t$ as:

$$\mathbf{B}_t = \{\mathbf{b}_t^k, o_t^k, \mathbf{c}_t^k, i_t^k\}_{k=1}^{K_0} = \mathcal{O}_t(\mathbf{P}_t). \tag{2}$$

The features of the objects are represented as $\mathbf{P}_{t,0} \in \mathbb{R}^{K_0 \times F_0}$.

Then we use a detection-based filter $\mathcal{F}_d$ to filter out bounding boxes with poor quality based on $o_t$,$\mathbf{c}_t$ and $i_t$. The filtering process involves three threshold judgments: using $o_t > \tau_{\text{obj}}$ to filter out proposals that may not be foreground objects; using $\max(\mathbf{c_t}) > \tau_{\text{cls}}$ to filter out proposals that are difficult to determine the category of the object; using $i_t > \tau_{\text{iou}}$ to filter out proposals that do not accurately locate the object. We sort the retained objects based on the global scores $s \times \max(\mathbf{c}) \times i$ and select the top $K_1$ objects with the highest product as the high-quality pseudo-labels $\{\mathbf{B}_{t,1}^i\}_{i=1}^{K_1}$. The process of selecting $K_1$ objects' features $\mathbf{P}_{t,1}$ and $\{\mathbf{B}_{t,1}^i\}_{i=1}^{K_1}$ is:

$$\begin{aligned}\mathbf{P}_{t,1} &= \mathbf{P}_{t,0}[\mathcal{F}_d(o_t, \mathbf{c}_t, i_t)], \\ \{\mathbf{B}_{t,1}^i\}_{i=1}^{K_1} &= \mathbf{B}_t[\mathcal{F}_d(o_t, \mathbf{c}_t, i_t)].\end{aligned} \tag{3}$$

**Pseudo label of captioning module.** The obtained high-quality object features $\mathbf{P}_{t,1}$ are then fed into the captioning module $\mathcal{C}_t$ of the teacher model to generate pseudo-captions for these $K_1$ objects $\{\mathbf{B}_{t,1}^i\}_{i=1}^{K_1}$. The generated captions are denoted as $\{\mathbf{L}_{t,1}^i\}_{i=1}^{K_1}$. However, many of these generated pseudo-captions are still of poor quality and cannot be used for cross-task training. Therefore, we adopt a simple strategy to filter out low-quality texts and only choose the high-quality text-object pair for later training. First, if the length of the pseudo-captions $\text{len}(\mathbf{L}_{t,1}^i) < \tau_{\text{cap}}$, the caption is likely to be several meaningless words and we filter is out. If the pseudo-captions do not contain any words or phrases describing the target object category, they should also be filtered out. When the number of unfiltered pseudo-captions reaches the preset number $K_2$, the generation process ends. If the remaining number of pseudo-captions is less than $K_2$, they are padded with empty texts. Note that these empty texts will not be used for the training of the student model. After this filtering stage, we obtain $K_2$ high-quality pseudo-captions $\{\mathbf{L}_{t,2}^i\}_{i=1}^{K_2}$ and their corresponding objects $\{\mathbf{B}_{t,2}^i\}_{i=1}^{K_2}$.

**Pseudo label of grounding module.** Previously obtained $K_2$ pseudo-captions may not necessarily provide sufficient information to locate the target object, or may not pinpoint a unique object. Therefore, we apply the grounding module $\mathcal{G}_t$ of the teacher model to further filter out pseudo-captions that cannot be accurately localized by the teacher model's grounding module. Specifically, we first input the $K_2$ pseudo-captions $\{\mathbf{L}_{t,2}^i\}_{i=1}^{K_2}$ and all unmatched $K_1$ objects detected by the detection module $\mathcal{O}_t$ into the grounding module $\mathcal{G}_t$ for matching, resulting in a set of confidence scores. The confidence score $\mathbf{s} \in \mathbb{R}^{K_1}$ for a certain pseudo-caption $\mathbf{L}_{t,2}^i$ represents its matching score with all $K_1$ objects. Then, we utilize the matched objects $\mathbf{B}_{t,2}^i$ of $K_2$ pseudo-captions as guidance, and extract scores corresponding to the objects $\mathbf{B}_{t,2}^i$ associated with the pseudo-description text, which reflects the ability of this pseudo-caption to localize the correct object. By comparing the scores, we sort and select the top $K_3$ pseudo-captions $\{\mathbf{L}_{t,3}^i\}_{i=1}^{K_3}$ with their corresponding objects $\{\mathbf{B}_{t,3}^i\}_{i=1}^{K_3}$.

Due to the different data augmentations applied to the input point clouds of the student and teacher, further pre-processing of the pseudo-labels is required before they can be utilized. Let $\mathcal{T}$ represent the combination of transformations applied to the student's input point clouds. Consequently, the pseudo-labels $b$ generated by the teacher for object positions must undergo the same transformations $b_T = \mathcal{T}(b)$. Furthermore, if the transformation $\mathcal{T}$ involves flipping along the y-axis, the pseudo-text descriptions generated by the teacher should also be modified accordingly, with the words "left" and "right" being interchanged. Following this processing, the student model can leverage these pseudo-labels for training.

## 3.5 Cross-Task Knowledge Transfer

Since the 3D grounding and captioning are complementary, simultaneously training these two tasks allows us to perform cross-task knowledge transfer: during the semi-supervised training stage, we transfer the learned knowledge between the two tasks with unlabeled data. Specifically, the predicted text obtained by the teacher model's captioning module $\mathcal{C}_t$ is first converted into tokens and encoded using GloVE [32]. Then, after pseudo-label filtering, this textual output serves as description text input for the student model's grounding module $\mathcal{G}_s$. In this manner, we transfer the knowledge learned by the captioning module of the teacher model to the grounding task on unlabeled data. Conversely, we employ the grounding module $\mathcal{G}_t$ of the teacher model to determine whether a piece of text can accurately locate the target object. This enables the selection of distinct object-text pairs for training the captioning module $\mathcal{C}_s$ of the student model to generate discriminative captions, thereby transferring the knowledge learned by grounding module of teacher model to the captioning task on unlabeled data.

To be specific, we utilize the pseudo-labels of detection module $\{\mathbf{B}_{t,1}^i\}_{i=1}^{K_1}$ and pseudo-label of object-text pair $\{\mathbf{L}_{t,3}^i\}_{i=1}^{K_3}$, $\{\mathbf{B}_{t,3}^i\}_{i=1}^{K_3}$ obtained from the teacher model to train the student model. The scene point cloud $\mathbf{P}_s$ is first inputted into the detection module $\mathcal{D}_s$ of the student model, yielding $K_0$ bounding boxes $\mathbf{B}_s$ of objects and their corresponding features $\mathbf{P}_s$ via:

$$\mathbf{B}_s, \mathbf{P}_s = \mathcal{D}_s(\mathbf{P}_s). \tag{4}$$

These features $\mathbf{P}_s$ are subsequently fed into both the grounding module $\mathcal{G}_s$ and captioning module $\mathcal{C}_s$ to accomplish the two tasks. For the grounding task, we query the object in $\mathbf{B}_s$ that matches a particular textual pseudo-label $\{\mathbf{L}_{t,3}^i\}_{i=1}^{K_3}$ best:

$$\mathbf{s}_{s,i} = \mathcal{G}_s(\mathbf{B}_s, \mathbf{L}_{t,3}^i), \tag{5}$$

where $\mathbf{s}_{s,i} \in \mathbb{R}^{K_0}$ represents the matching score between the given text and the $K_0$ objects $\mathbf{B}_s$. We supervise it using corresponding object pseudo-labels $\{\mathbf{B}_{t,3}^i\}_{i=1}^{K_3}$. For the captioning task, we identify the object in $\mathbf{B}_s$ with the highest IoU with a given object pseudo-label $\{\mathbf{B}_{t,3}^i\}_{i=1}^{K_3}$ and generate a textual description for it:

$$\mathbf{L}_{s,i} = \mathcal{C}_s(\mathbf{B}_s, \mathbf{B}_{t,3}^i). \tag{6}$$

We supervise it with the pseudo-label $\{\mathbf{L}_{t,3}^i\}_{i=1}^{K_3}$ associated with that object. Additionally, we also employ detection pseudo-labels $\{\mathbf{B}_{t,1}^i\}_{i=1}^{K_1}$ to supervise the results of object detection $\mathbf{B}_s$ in the student branch.

---

**Algorithm 1:** Cross-Task Teacher-Student Framework

**Input:** Labeled data $D_l = \{\mathbf{P}_i^l, \mathbf{L}_i^l, \mathbf{B}_i^l\}_{i=1}^{N_s}$, unlabeled data
$D_u = \{\mathbf{P}_i^u\}_{i=1}^{N_u}$

1   Initialize teacher: $[O_t, C_t, \mathcal{G}_t] \leftarrow$ random parameters;

2   **for** $j < E_1$ **do**    // Train teacher with labeled data

3      Compute labeled loss $\mathcal{L}_l$ by Equation (10);

4      Update $[O_t, C_t, \mathcal{G}_t]$ by Equation (1);

5   **end for**

6   share weights from teacher model to student model
    $[O_s, C_s, \mathcal{G}_s] \leftarrow [O_t, C_t, \mathcal{G}_t]$;

7   **for** $j < E_2$ **do**    // Train student with pseudo-labels

8      Generate pseudo-labels $\{\mathbf{B}_{t,1}^i\}_{i=1}^{K_1}, \{\mathbf{L}_{t,3}^i\}_{i=1}^{K_3}, \{\mathbf{B}_{t,3}^i\}_{i=1}^{K_3}$
     following the Sec.3.4;

9      Compute unlabeled loss $\mathcal{L}_u$ with pseudo-labels by
     Equation (11);

10     Compute overall loss $\mathcal{L}$ by Equation (7);

11     Update both teacher and student models $[O_s, C_s, \mathcal{G}_s]$,
     $[O_t, C_t, \mathcal{G}_t]$ by Equation (8);

12   **end for**

**Output:** Student model parameters $[O_s, C_s, \mathcal{G}_s]$

---

The overall loss function can be expressed as:

$$\mathcal{L} = \mathcal{L}_l + \lambda_u \mathcal{L}_u, \tag{7}$$

where $\mathcal{L}_l, \mathcal{L}_u$ are the loss functions for labeled and unlabeled data (illustrated in Sec.3.6), and $\lambda_u$ is the weight coefficient. After obtaining the loss function $\mathcal{L}$, the parameters of the student model $[O_s, C_s, \mathcal{G}_s]$ are optimized through gradient descent, and the parameters of the teacher model $[O_t, C_t, \mathcal{G}_t]$ are optimized through exponential moving average (EMA) by the following equation:

$$[O_s, C_s, \mathcal{G}_s]^j = [O_s, C_s, \mathcal{G}_s]^{j-1} + \gamma \frac{\partial \mathcal{L}}{\partial [O_s, C_s, \mathcal{G}_s]^{j-1}},$$
$$[O_t, C_t, \mathcal{G}_t]^j = \alpha[O_t, C_t, \mathcal{G}_t]^{j-1} + (1-\alpha)[O_s, C_s, \mathcal{G}_s]^j, \tag{8}$$

where $\alpha$ is an EMA smoothing coefficient, $\gamma$ is the learning rate and $j$ is the training step index. The whole semi-supervised training algorithmic procedure pseudo-code is shown in Algorithm 1.

## 3.6   Training Losses

**Labeled data loss.** We utilize ground truth to compute the labeled loss $\mathcal{L}_l$. Given the integration of both visual grounding and dense captioning tasks, a combination of multiple loss functions is required to supervise the training process. The first component of the loss function is the detection loss $L_{l,\text{det}}$, which consists of the losses for objectness score, object category prediction, IoU prediction, and bounding box prediction, denoted as $L_{l,\text{obj}}, L_{l,\text{sem}}, L_{l,\text{iou}}$, and $L_{l,\text{box}}$:

$$L_{l,\text{det}} = \lambda_1 L_{l,\text{obj}} + \lambda_2 L_{l,\text{sem}} + \lambda_3 L_{l,\text{iou}} + \lambda_4 L_{l,\text{box}}. \tag{9}$$

The second component of the loss function is the grounding task loss $L_{l,\text{grd}}$. Similar to ScanRefer [8], it is obtained by computing the loss between the confidence score and the one-hot target label obtained from the ground truth. The third component is the captioning task loss $L_{l,\text{cap}}$, representing the cross-entropy loss for

token prediction [41, 47]. The overall loss function for labeled data is the weighted sum of these three components:

$$\mathcal{L}_l = L_{l,\text{det}} + \lambda_5 L_{l,\text{grd}} + \lambda_6 L_{l,\text{cap}}. \tag{10}$$

**Unlabeled data loss.** We compute unlabeled data loss $\mathcal{L}_u$ using pseudo-labels generated by the teacher model. The computation method is generally similar to that of labeled data, but for unlabeled data, we no longer supervise the prediction of IoU and objectness score [42], only using them for pseudo-label filtering:

$$L_{u,\text{det}} = \lambda_2 L_{u,\text{sem}} + \lambda_4 L_{u,\text{box}},$$
$$\mathcal{L}_u = L_{u,\text{det}} + \lambda_5 L_{u,\text{grd}} + \lambda_6 L_{u,\text{cap}}. \tag{11}$$

## 4   EXPERIMENTS

### 4.1   Datasets and Evaluation Metric

For the 3D grounding task, we conducted experiments on widely used ScanRefer dataset [8] and Nr3D dataset [1]. ScanRefer contains a total of 51,583 textual descriptions corresponding to the objects provided in 806 scanned scenes from the ScanNet [14] dataset. We utilize the metric Acc@kIoU, where "k" represents the minimum threshold for the IoU between the predicted bounding box and the ground truth. Following previous works [8, 49], we set "k" to 0.25 and 0.5 for our experiments. To provide a comprehensive analysis, we presented the results separately for the "unique" and "multiple" subsets. The "unique" subset refers to scenes where there is only one object of the same category as the target object, while the "multiple" subset includes scenes with multiple objects of the same category. Nr3D [1] provides 41.5K textual descriptions for scenes in ScanNet. We evaluate the effectiveness of our method on NR3D using the same metrics. For the 3D captioning task, we followed the approach of Scan2Cap [11] to handle the description text of the ScanRefer dataset. Texts exceeding 30 tokens were truncated to 30 tokens, with [SOS] and [EOS] tokens added at the beginning and end, respectively. To evaluate the performance of the method on captioning tasks, we adopted the common used m@kIoU as metric, where m represents captioning metrics CiDEr [40], BLEU-4 [31], METEOR [4], and ROUGE-L [25]. These metrics jointly measure the quality of both object detection and text generation.

### 4.2   Implementation Details

We utilized four NVIDIA RTX3090 GPUs for training. During the warm-up and pseudo-label learning stages, we trained for 200 epochs (i.e., $E1, E2 = 200$) each with a batch size of 8. For each stage, we used an AdamW [28] optimizer with a cosine learning rate decay strategy. The initial learning rate for warm-up stage was set to 5e-4, while for the pseudo-label learning stage, it was set to 2.5e-4. In our experiments, the thresholds $\tau_{\text{obj}}, \tau_{\text{cls}}, \tau_{\text{iou}}$, and $\tau_{\text{cap}}$ used in the pseudo-label filtering stage were set to 0.5, 0.5, 0.15, and 5 respectively. The remaining numbers of pseudo-labels after filtering for each stage $K_1$, $K_2$, and $K_3$, were set to 128, 16, and 8 respectively. As for the loss function, the coefficients $\lambda_1, \lambda_2, \lambda_3, \lambda_4$, $\lambda_5, \lambda_6$ and $\lambda_u$ are set to 0.1, 0.1, 0.5, 20, 0.2, 0.3, 1 respectively. The EMA smoothing coefficient $\alpha$ is set to 0.9995.

**Table 1: Comparison on ScanRefer dataset, where our semi-supervised (SS) method performs better than most fully-supervised (FS) methods and all weakly-supervised (WS) methods.** *Note that, although WS setting does not rely on bbox labels, it requires query labels of all scenes. Compared to FS and WS settings, our SS setting is more challenging as we only require limited annotations.*

| Method | Modality | Setting | Annotation | | | Unique | | Multiple | | Overall | |
|---|---|---|---|---|---|---|---|---|---|---|---|
| | | | query | bbox | Percent. | 0.25 | 0.5 | 0.25 | 0.5 | 0.25 | 0.5 |
| ScanRefer [8] | 3D+2D | FS | ✓ | ✓ | 100% | 76.33 | 53.51 | 32.73 | 21.11 | 41.19 | 27.40 |
| Referit3D [1] | 3D | FS | ✓ | ✓ | 100% | 53.80 | 37.50 | 21.00 | 12.80 | 26.40 | 16.90 |
| TGNN [20] | 3D | FS | ✓ | ✓ | 100% | 68.61 | 56.80 | 29.84 | 23.18 | 37.37 | 29.70 |
| InstanceRefer [51] | 3D | FS | ✓ | ✓ | 100% | 77.45 | 66.83 | 31.27 | 24.77 | 40.23 | 32.93 |
| SAT [49] | 3D+2D | FS | ✓ | ✓ | 100% | 73.21 | 50.83 | 37.64 | 25.16 | 44.53 | 30.14 |
| 3DVG-Transformer [52] | 3D+2D | FS | ✓ | ✓ | 100% | 81.93 | 60.64 | 39.30 | 28.42 | 47.57 | 34.67 |
| 3DJCG [7] | 3D+2D | FS | ✓ | ✓ | 100% | 83.47 | 64.34 | 41.39 | 30.82 | 49.56 | 37.33 |
| EDA [45] | 3D | FS | ✓ | ✓ | 100% | 85.76 | 68.57 | 49.13 | 37.64 | 54.59 | 42.26 |
| WS-3DVG [44] | 3D | WS | ✓ | ✗ | 100% | - | - | - | - | 10.43 | 6.37 |
| 3D-VLA [48] | 3D | WS | ✓ | ✗ | 100% | 72.95 | 62.17 | 22.77 | 17.94 | 32.51 | 26.53 |
| Ours | 3D | SS | partial | partial | 10% | 76.75 | 58.27 | 38.60 | 28.98 | 46.00 | 34.67 |
| | 3D+2D | SS | partial | partial | 10% | 82.87 | 61.79 | 39.57 | 28.89 | 47.97 | 35.28 |

**Table 2: Re-implementing all methods trained on solely 10% labeled data. Experiments are conducted on ScanRefer.**

| Method | Modality | Overall | |
|---|---|---|---|
| | | Acc@0.25 | Acc@0.5 |
| ScanRefer [8] | 3D+2D | 29.21 | 12.07 |
| Referit3D [1] | 3D | 12.35 | 9.77 |
| TGNN [20] | 3D | 9.08 | 7.64 |
| InstanceRefer [51] | 3D | 33.37 | 27.03 |
| SAT [49] | 3D+2D | 29.51 | 12.24 |
| 3DVG-Transfromer [52] | 3D+2D | 30.54 | 13.01 |
| 3DJCG [7] | 3D+2D | 40.88 | 27.42 |
| EDA [45] | 3D | 39.53 | 23.62 |
| Ours | 3D | 46.00 | 34.67 |
| | 3D+2D | 47.97 | 35.28 |

**Table 3: Results of 3D captioning task on Scan2Cap.**

| Method | label | C@0.5 | B-4@0.5 | M@0.5 | R@0.5 |
|---|---|---|---|---|---|
| VoteNetRetr [33] | 100% | 10.18 | 13.38 | 17.14 | 33.22 |
| Scan2Cap [11] | 100% | 39.08 | 23.32 | 21.97 | 44.48 |
| SpaCap3d [43] | 100% | 42.76 | 25.38 | 22.84 | 45.66 |
| 3DJCG [7] | 100% | 49.48 | 31.03 | 24.22 | 50.80 |
| Ours | 10% | 33.00 | 28.37 | 22.50 | 48.82 |

transfer. Performance comparison on more datasets like Nr3D can be found in our **Supplementary files**.

**3D captioning results.** We also compare the performance of our generated captions with several captioning methods in Table 3. It can be observed that our approach with only 10% labeled data achieves comparable results to many methods utilizing 100% labeled data. This indicates that our method also gains the ability to learn captioning task from unlabeled data through knowledge transfer.

## 4.3 Comparison with SOTA Model

**3D grounding results.** In Table 1, we present a comparison of our method with previous fully supervised and weakly supervised approaches for the 3D grounding task on the ScanRefer dataset. From the table, it can be observed that with only 10% of labeled data, our method achieves better results than many fully supervised methods that utilize all labeled data, although slightly inferior to some state-of-the-art methods. This demonstrates the effectiveness of our approach. The table also showcases two weakly supervised methods, which utilize all textual annotations but lack corresponding bounding boxes. Compared to them, our approach solely utilizes a small amount of fully labeled data along with a large amount of unlabeled scene point cloud data. Furthermore, the 3D-VLA [48] method heavily relies on pre-trained models in the 2D domain. Therefore, directly comparing these weakly-supervised methods to us is not feasible. However, we still outperforms their performance a lot. To fairly compare the existing methods in the same semi-supervised setting, we also re-implement previous approaches by using only 10% labeled data. As shown in Table 2, our method significantly outperforms all previous methods when using the same amount of labeled data. This clearly validates the ability of our approach to acquire knowledge for 3D visual grounding from unannotated scenes through pseudo-label learning and knowledge

## 4.4 Ablation Study

**Ablation study on varying proportions of labeled data.** We conducted experiments under various proportions of labeled data, as shown in Figure 4. The results indicate that our approach is capable of learning useful information from a small amount of labeled data (e.g., 3%). As the proportion of labeled data increases, our method is able to generate better pseudo-labels, thereby enhancing the final experimental outcomes. Furthermore, comparisons with the baseline method 3DJCG under different annotation proportions reveal significant and consistent enhancements across different proportions of labeled data, demonstrating the ability of our method in learning from unlabeled data.

**Ablation study on pseudo-label filtering step.** We propose three pseudo-label filtering schemes based on detection, captioning, and grounding. To assess the effectiveness of these filtering schemes, we conducted ablation study and presented the results in Table 4. The first row of the table represents the results of semi-supervised training using all pseudo-labels generated by the teacher model, while the last row corresponds to our proposed method. From the table, we can conclude that our designed filtering schemes at

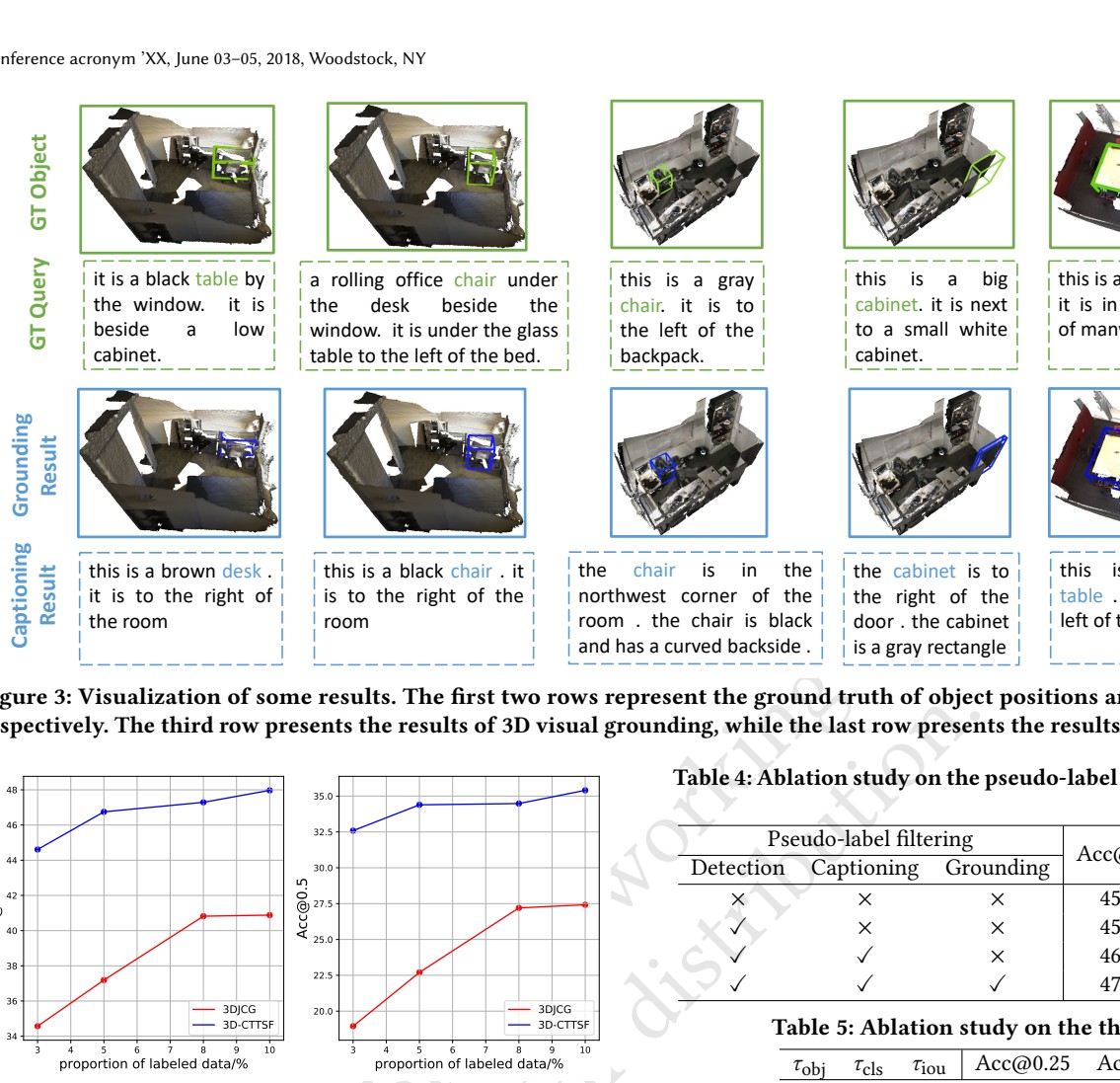

**Figure 3: Visualization of some results. The first two rows represent the ground truth of object positions and the query text, respectively. The third row presents the results of 3D visual grounding, while the last row presents the results of 3D captioning.**

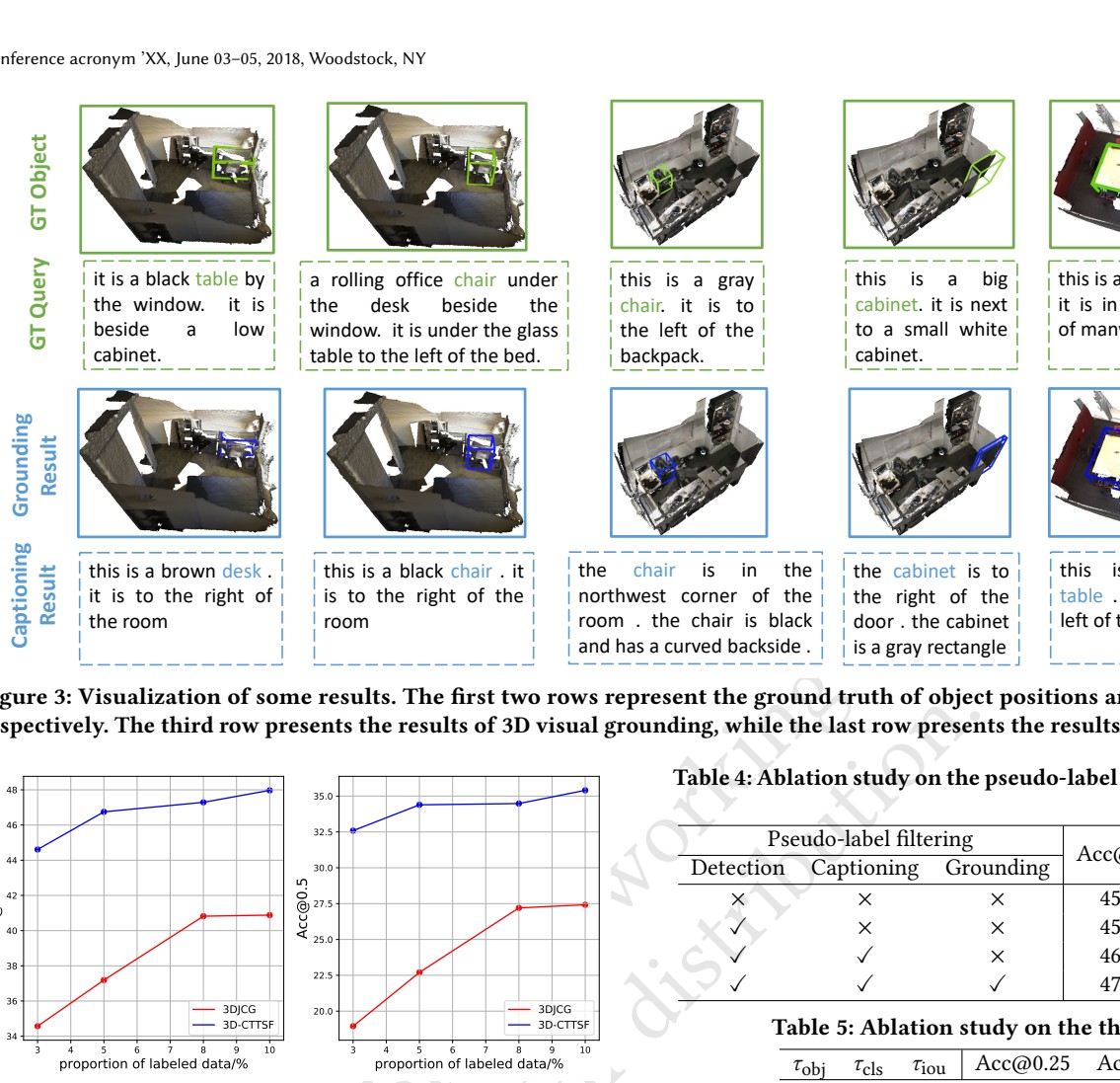

**Figure 4: Ablation study on proportions of labeled data.**

the detection, captioning, and grounding levels all contribute to improving the results of semi-supervised learning.

**Ablation study on the filtering thresholds.** We conducted further ablation studies on the thresholds used in the pseudo-label filtering step. The experimental results for the $\tau_{obj}$, $\tau_{cls}$, and $\tau_{iou}$ are shown in Table 5. The table shows that both excessively large or small thresholds can compromise the effectiveness of the method. Excessively large thresholds may result in an insufficient number of pseudo-labels, thus preventing the student model from receiving adequate training. Conversely, excessively small thresholds may lead to the utilization of some low-quality pseudo-labels for training, thereby compromising the training results.

More ablation study can be found in our **Supplementary files**.

### 4.5 Visualization

In Figure 3, we present visualizations of some output results obtained using our approach with 10% annotated data. It can be observed that our method is capable of performing both 3D visual grounding and dense captioning tasks with a minimal amount of annotated data. It is noteworthy that the generated captions uniquely

**Table 4: Ablation study on the pseudo-label filtering schemes.**

| Pseudo-label filtering | | | Acc@0.25 | Acc@0.5 |
|---|---|---|---|---|
| Detection | Captioning | Grounding | | |
| ✗ | ✗ | ✗ | 45.70 | 30.48 |
| ✓ | ✗ | ✗ | 45.77 | 31.04 |
| ✓ | ✓ | ✗ | 46.66 | 33.40 |
| ✓ | ✓ | ✓ | 47.97 | 35.28 |

**Table 5: Ablation study on the threshold.**

| $\tau_{obj}$ | $\tau_{cls}$ | $\tau_{iou}$ | Acc@0.25 | Acc@0.5 |
|---|---|---|---|---|
| 0.5 | 0.5 | 0.15 | 47.97 | 35.28 |
| 0.1 | 0.5 | 0.15 | 47.25 | 34.81 |
| 0.9 | 0.5 | 0.15 | 47.85 | 34.96 |
| 0.5 | 0.1 | 0.15 | 47.09 | 34.11 |
| 0.5 | 0.9 | 0.15 | 47.31 | 33.71 |
| 0.5 | 0.5 | 0.25 | 47.62 | 35.63 |
| 0.5 | 0.5 | 0.5 | 47.66 | 35.18 |

describe the target object based on various features, demonstrating knowledge transfer from the grounding task.

### 5 CONCLUSION

In this paper, we propose to address an important yet challenging task, *i.e.*, semi-supervised 3D grounding. Since most scene data has no annotation, we explore additionally training a 3D captioning module to assist the grounding learning. To this end, we develop a joint 3D grounding and captioning backbone with a teacher-student framework design to tackle this semi-supervised learning. Cross-task pseudo-label generation and knowledge transfer strategies are further introduced to improve pseudo-label learning and updating. Extensive experiments demonstrate the proposed framework bridges the gap between semi-supervised learning and fully-supervised learning on various datasets.

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

Received 20 February 2007; revised 12 March 2009; accepted 5 June 2009

