# OpenReview forum: "Cross-Task Knowledge Transfer for Semi-supervised Joint 3D Grounding and Captioning"
_acmmm.org/ACMMM/2024/Conference — MM2024 Oral_

### Official Review · Reviewer_ik5c · 2024-05-05

**Rating:** 5
**Confidence:** 3

**Summary:**

The paper focuses on learning a 3D visual grounding model with minimal annotations by introducing a semi-supervised setting for the 3D visual grounding task. It proposes leveraging knowledge transfer from the related task of 3D captioning to improve the accuracy of 3D scene understanding. By training these tasks iteratively with unlabeled data, the authors aim to enhance object and text contexts within 3D scenes.

**Strengths:**

Innovative Approach: The paper introduces a novel semi-supervised setting for 3D visual grounding, addressing the challenge of limited annotated data.
Knowledge Transfer: Leveraging knowledge transfer from the 3D captioning task demonstrates a creative way to improve the performance of 3D scene understanding.
Iterative Training: The iterative training approach with unlabeled data shows promise in enhancing the object and text contexts within 3D scenes.

**Limitations:**

One specific limitation of the paper is the lack of information regarding the scalability and generalizability of the proposed 3D Cross-Task Teacher-Student Framework (3D-CTTSF). While the document discusses the effectiveness of the framework in a semi-supervised setting with limited annotations, it does not provide insights into how well the approach would perform across different datasets, varying levels of complexity in 3D scenes, or in real-world applications beyond the scope of the study. Understanding the framework's scalability and generalizability is crucial for assessing its potential impact and practical utility in diverse multimedia understanding tasks.

**Suitability:**

3

---

### Official Review · Reviewer_6CgH · 2024-05-22

**Rating:** 5
**Confidence:** 3

**Summary:**

The paper proposed a semi-supervised training strategy for 3D visual grounding utilizing very few labels. The method relies on alternate 3D captioning and Grounding using a teacher-student training framework. Teacher warmed up to small scale labelled data;
Further Teacher produces pseudo object proposals and pseudo labels to train student.

**Strengths:**

The proposed method uses semi-supervised setting inside a teacher-student framework for learning 3D visual grounding. This setting is novel for 3D visual grounding task. The combination of captioning and grounding for 3D visual has been explored before as in [1]. In this work, it is extended to teacher-student training where the pseudo labels from teacher is used to train student on unlabelled point cloud data. The object and caption pseudo labels from teacher is carefully filtered to ensure quality of labels. Moreover data augmentation is differently applied to teacher and student to enable effective learning of student network.

The proposed work has some novelties compared to existing literature in 3D visual grounding
-	It addresses the problem of 3D visual grounding in a semi-supervised setting utilizing teacher pseudo labels to train student network
-	The design allows cross task knowledge transfer while learning  captioning and grounding   with pseudo labels .

The experimental results are performed satisfactorily with adequate comparisons to prove the usefulness of proposed method.

[1] 3DJCG: A Unified Framework for Joint Dense Captioning and Visual Grounding on 3D Point Clouds, CVPR 2022.

**Limitations:**

A good pseudo label caption should ideally describe an object in the context of its  local neighborhood to differentiate it from similar objects in the scene. This characteristic is not coming out in the design. The ‘goodness’ of a pseudo label caption is decided by the ‘easiness’  to ground using the grounding module. However ,this can result in producing more descriptive captions of unique objects neglecting spatial layouts. It is not clear how this will be handled in the work.

**Suitability:**

3

---

### Official Review · Reviewer_kKUG · 2024-05-24

**Rating:** 4
**Confidence:** 3

**Summary:**

In this paper, authors propose a 3D cross-task teacher-student framework for joint 3D grounding and captioning in the semi-supervised setting. Experimental results on benchmarks prove the effectiveness of proposed framework.

**Strengths:**

1.The written is clear and easy to follow.
2.The expansion to semi-supervised setting is reasonable and results seem promising.
3.high completion quality.

**Limitations:**

1.The idea of leveraging cross-task knowledge transfer to enhance performance on the current task has been proven effective by previous works, even under semi-supervised learning settings.

2.The in-distribution setting based on ScanRefer's grounding annotations plus captioning annotations is clearly a straightforward and simple setup. Have the authors considered using a cross-domain 3D scene setup, aiding the in-distribution (ID) tasks with out-of-distribution (OOD) datasets, such as using outdoor scene point clouds and captioning to assist indoor scenes, or vice versa?

3.Given the in-distribution (ID) setting, the semi-supervised setting of this paper seems to have limited significance. If high-quality pseudo-labeling using cross-domain scene data could be achieved to further enhance performance, it would greatly improve the reliability of the method.

4.In the process of pseudo-labeling, could the label distribution of categories be taken into consideration? For instance, in ScanRefer, chairs are the most frequently occurring objects, while bathtubs are rare. After training with only 10% of the data, the model's generalization for bathtubs might be very limited. Therefore, the quality of pseudo-labels for these less frequently occurring objects could also be expected to be low on unlabeled data. Could you provide more insights?

5. Figure 2 is too complex and not easy to understand.

**Suitability:**

3

---

### Official Review · Reviewer_Yn85 · 2024-05-27

**Rating:** 4
**Confidence:** 3

**Summary:**

The main content of the paper is about a novel approach called "Cross-Task Knowledge Transfer for Semi-supervised Joint3D Grounding and Captioning". The paper proposes a framework that combines 3D visual grounding and dense captioning tasks in a semi-supervised learning setting. The framework utilizes a teacher-student architecture to transfer knowledge between the two tasks and generate pseudo-labels for unlabeled data. Experimental results demonstrate that the proposed framework achieves promising performance on various datasets.

**Strengths:**

It is novelty that bridges the gap between semi-supervised learning and fully-supervised methods for the joint tasks of 3D grounding and captioning. The proposed method is extensively evaluated on various datasets for both grounding and captioning tasks. The evaluation results demonstrate the effectiveness of the 3D-CTTSF in semi-supervised learning, showing improved performance compared to previous methods.

**Limitations:**

There is also another paper D3Net, which is also a paper to do 3D Dense Captioning and visual grounding. What are the differences between this paper and the D3Net? Perhaps, this paper is not the first work.

**Suitability:**

2

---

### Meta-Review · Area_Chair_EvW4 · 2024-06-27

**Recommendation:** Accept (Oral)
**Confidence:** 4

**Metareview:**

The paper introduces an innovative framework, named the 3D Cross-Task Teacher-Student Framework (3D-CTTSF), which integrates 3D visual grounding with dense captioning in a semi-supervised learning environment. This approach is novel for the 3D visual grounding task. The experimental results on established benchmarks validate the efficacy of the proposed framework.

The paper receives the initial rating of two borderline accept and two weak accept. After the rebuttal, all reviewers are positive towards the paper, with the final scores of 5, 5, 5, 6, acknowledging the method’s novelty and effectiveness. The AC agrees with the rating and recommends accepting this paper.